# ANALYSIS OF DIFFERENTIALLY PRIVATE SYNTHETIC DATA: A MEASUREMENT ERROR APPROACH

## ABSTRACT

Differential private (DP) synthetic datasets have been receiving significant attention from academia, industry, and government. However, little is known about how to perform statistical inference using DP synthetic datasets. Naive approaches that do not take into account the induced uncertainty due to the DP mechanism will result in biased estimators and invalid inferences. In this paper, we present a class of MLE-based easy-to-implement bias-corrected DP estimators with valid asymptotic confidence intervals (CI) for parameters in regression settings, by establishing the connection between additive DP mechanisms and measurement error models. Our simulation shows that our estimator has comparable performance to the widely used sufficient statistic perturbation (SSP) algorithm in some scenarios but with the advantage of releasing a synthetic dataset and obtaining statistically valid asymptotic CIs, which can achieve better coverage when compared to the naive CIs obtained by ignoring the DP mechanism.

## 1 INTRODUCTION

Differential privacy (DP) is a mathematically rigorous definition that quantifies privacy risk. It builds on the idea of releasing "privacy-protected" query results, such as summary statistics, using randomized responses. In recent years, the use of differential privacy has quickly gathered popularity as it promises that there will be no additional privacy harm for any individual whether the said individual's data belongs in a private dataset or not, and therefore encourage data sharing.

One important characteristic of DP is its composition property (Dwork & Roth, 2014). That is, to avoid the scenario where the same analysis is rerun and averaging away the noises from the randomized responses, the composition property indicates that running the same analysis twice will have double the amount of privacy risk as running the analysis once. The data provider often set a total amount of privacy risk/budget allowed, commonly referred to as the privacy budget, and each analysis from researchers uses a portion of the privacy budget. Once the total privacy budget is exhausted, any new analysis would not be possible unless the data provider decides to increase the total privacy budget and thus take on more privacy risk. This could be problematic as it limits the number of analyses that researchers can run, which can result in the dataset not being fully explored. In consequence, it diminishes the probability of serendipitous discovery and amplifies the odds of being tricked by unanticipated data problems (Evans et al., Working Paper).

To address the problem above, various methods of releasing differentially private synthetic datasets (Liu (2016); Bowen & Liu (2020); Gambs et al. (2021)) have been proposed. Using the post-processing property of DP, any analysis on the DP synthetic dataset will be differentially private without the additional cost of privacy budget. Therefore, by releasing DP synthetic dataset, it circumvents the problem of running out of privacy budget. Here we will mention a few notable methods of generating DP synthetic datasets. In general, the methods of generating DP datasets can be categorized into the non-parametric method and the parametric method. For the non-parametric methods, the DP dataset is constructed based on the empirical distribution of the data. The simplest approach would be directly adding Laplace or Gaussian noises to the confidential dataset. For the parametric methods, the DP dataset is constructed based on a parametric distribution/model of the data. Using the robust and flexible model of vine copula, Gambs et al. (2021) draw the DP synthetic dataset from the DP trained vine copula model. From the Bayesian perspective, Liu (2016) proposes generating DP synthetic dataset by drawing samples from the DP version of the posterior predictive

distribution. For a more comprehensive overview of different DP dataset generation methods, refer to Bowen & Liu (2020).

Dwork & Roth (2014) characterizes differential privacy as a definition of privacy tailored to the problem of privacy-preserving data analysis. However, for a statistician, the goal of statistical inference is often as important as data analysis. Under the framework of differential privacy, the methods for making statistical inferences are under-explored. Fortunately, the interest in statistical inference under differential privacy has been rising recently including the works like Sheffet (2017) and Barrientos et al. (2019). To make statistical inferences, statistical models need to be specified. With different differential privacy algorithms, it comes different statistical models. It turns out that additive mechanisms, like the Laplace mechanism or Gaussian mechanism, give statistical models that are naturally related to the measurement error models. In other words, each additive mechanism can be viewed as some variation of the measurement error model, and therefore, the tools from the measurement error models can be used to make inferences in the differential privacy setting.

In this paper, we generate DP synthetic dataset by adding DP noises directly to the confidential dataset through the Gaussian mechanism. We choose this method due to its simplicity, and more importantly, it allows us to establish the connection to the theory of measurement error as we will see in section 3.1. Using the established tool in the theory of measurement error, we then derive an MLE-based DP bias-corrected estimator and an asymptotic confidence interval for our parameter of interest. Therefore, by establishing a connection to measurement error, we will be able to develop statistical inference under the differential privacy setting. To demonstrate the usefulness of this connection, we study statistical inference under the linear regression setting while preserving differential privacy. In particular, we derive DP consistent estimator and asymptotic confidence interval for the regression coefficient.

**Related work**   As one of the most common statistical models, linear regression has been studied before in differential privacy literature. One of the widely used methods for obtaining a DP estimator for the regression coefficient is through the perturbation of sufficient statistics (Dwork et al., 2014; Sheffet, 2017; Wang, 2018). It's commonly used due to its simplicity and is closely related to the classical ordinary least square method. Motivated by Dwork & Lei (2009), Alabi et al. (2022) shows that algorithms based on a robust estimator, such as a median-based estimator, perform better compared to the classical ordinary least square estimator on small sample cases. Similar to our work, Charest & Nombo (2020) uses simulation extrapolation (SIMEX), a technique from the literature on measurement error (Carroll et al. (2006)), to obtain a DP estimator for the regression coefficient. However, what differs from our work is that there is no mention of constructing confidence intervals in Charest & Nombo (2020). Agarwal et al. (2021); Agarwal & Singh (2021) also mentioned the connection between the measurement error model and differentially private mechanism. Differing from our work, Agarwal et al. (2021) focused on the setting where only covariates are perturbed with differentially private noises and on the goal of learning a predictive linear model using principal component regression. Similar to our work, Agarwal & Singh (2021) use the connection to make inferences on the regression coefficient under a more general and less structured setting, but the methodology is much more involved compared to our more simplistic approach. Lastly, Evans & King (2022) also uses the connection to obtain a consistent estimator of the regression coefficient, but without any mention of the confidence interval.

Using the Johnson-Lindenstrauss transform (Blocki et al., 2012), Sheffet (2017) studies DP ordinary least square estimator and derived DP asymptotic confidence intervals for the regression coefficients. Differing from the additive DP noises used in our work, a random projection could potentially limit the usefulness of the synthetic dataset for other types of analysis. Instead of obtaining confidence intervals for the regression coefficient, Barrientos et al. (2019) studies the DP hypothesis testing for the regression coefficient by perturbing the t-statistic. However, since the approach achieves DP through randomizing the t-statistics, each hypothesis testing will cost a portion of the total privacy budget and the total privacy budget can be exhausted quickly.

Lastly, from the Bayesian perspective, Bernstein & Sheldon (2019) studies the DP Bayesian linear regression, which requires a prior distribution for the regression coefficient, through releasing private sufficient statistics and thus is imperilled to the problem of privacy budget running out as described above.

**Structure of the paper**  In section 2, we state the necessary concepts related to differential privacy and measurement error. In section 3, we establish the connection between differential privacy and measurement error and working under regression setting, we derive DP consistent estimator and DP asymptotic confidence interval for regression coefficient $\beta$ using the tool from measurement error framework. In section 4, we conduct a simulation to examine the performance of our DP estimator against the widely used sufficient statistics perturbation method (SSP) and show that the performance of our estimator is comparable to SSP estimator in some scenarios while being outperformed in others. Furthermore, we look at the coverage of our DP confidence interval compared with the naive CIs obtained from ignoring the DP noises, and demonstrate the issue with naive inference as not only are the naive CIs centred at the wrong value, but they also have shorter length than would be obtained with the true data (Carroll et al., 2006).

## 2 PRELIMINARIES

### 2.1 DIFFERENTIAL PRIVACY

We begin by going through some basics regarding differential privacy. The central idea around differential privacy is that it gives the assurance that any sequence of query responses is "essentially" equally likely to occur, independent of the presence or absence of any individual (Dwork & Roth (2014)). Naturally, the most basic concept in differential privacy is the notion of neighbouring datasets where two datasets differ by one individual record.

**Definition 2.1** (Neighboring datasets). *Two datasets of the same dimension (same numbers of columns and rows) are called neighbouring datasets if they only differ in exactly one row/individual record.*

In this paper, we are only concerned with approximate differential privacy, which is a natural relaxation of the original definition of $\varepsilon$-differential privacy.

**Definition 2.2** (Approximate differential privacy (Dwork et al., 2006a;b)). *A randomized algorithm $\mathcal{M}$ is $(\varepsilon, \delta)$-differentially private if for all (measurable) set $\mathcal{S}$ and for all neighboring datasets $\mathbf{X}$ and $\mathbf{X}'$,*

$$\mathbb{P}(\mathcal{M}(\mathbf{X}) \in \mathcal{S}) \leq \exp(\varepsilon)\mathbb{P}(\mathcal{M}(\mathbf{X}') \in \mathcal{S}) + \delta$$

One of the most important properties of differential privacy is its immunity to post-processing. That is, without any additional information on the confidential dataset, it's impossible to make a function of the output of a differentially private algorithm $\mathcal{M}$ any less differentially private. More precisely,

**Proposition 2.1** (Post-processing property (Dwork & Roth, 2014)). *Let $\mathcal{M}$ be randomized algorithm that is $(\varepsilon, \delta)$-differentially private. Let $f$ be an arbitrary randomized mapping with its domain within the range of $\mathcal{M}$. Then $f \circ \mathcal{M}$ is $(\varepsilon, \delta)$-differentially private.*

Another fundamental notion in differential privacy is the idea of a query. A *query* is a function to be applied to the dataset (Dwork & Roth (2014)). Naturally, to achieve the same degree of privacy protection, different queries will likely require a different amount of noise perturbation. To quantify this, we need the concept of *sensitivity*:

**Definition 2.3** ($l_2$ sensitivity). *The $l_2$ sensitivity of a query function $f$ is defined as $\Delta_f = \max_{\mathbf{X},\mathbf{X}'} \|f(\mathbf{X}) - f(\mathbf{X}')\|_2$ where the $\max$ is taken over all possible pair of neighboring datasets $\mathbf{X}$ and $\mathbf{X}'$.*

$(\varepsilon, \delta)$-differential privacy can be achieved through the application of *Gaussian mechanism*,

**Definition 2.4** (Analytic Gaussian Mechanism (Balle & Wang, 2018)). *Let $f : \mathbb{X} \to \mathbb{R}^d$ be a function with global $L_2$ sensitivity $\Delta$. For any $\varepsilon \geq 0$ and $\delta \in [0, 1]$, the Gaussian output perturbation mechanism $M(x) = f(x) + Z$ with $Z \sim \mathcal{N}\left(0, \sigma^2 I\right)$ is $(\varepsilon, \delta)$-DP if and only if*

$$\Phi\left(\frac{\Delta}{2\sigma} - \frac{\varepsilon\sigma}{\Delta}\right) - e^\varepsilon \Phi\left(-\frac{\Delta}{2\sigma} - \frac{\varepsilon\sigma}{\Delta}\right) \leq \delta$$

*where $\Phi$ is the cumulative distribution function for a standard normal random variable.*

## 2.2 MEASUREMENT ERROR MODEL

In simplest terms, measurement error problems can be described as the problem of making inferences about a statistical model in terms of a variable $Z$ that is not directly observable. Instead, a surrogate variable $W$ of $Z$ is observed, and inference must be made through $W$ instead. The statistical models and inference methods are called measurement error models (Stefanski, 2000).

A measurement error model consists of two parts, the first is the *error structure* relating the surrogate $W$ to the truth $Z$, and the second is *data structure* of true variable $Z$ (Carroll et al., 2006). As an example, consider the following measurement error model,

$$W = X + U \tag{1a}$$
$$Y = g(X; \beta) + q \tag{1b}$$

where the $U$ is assumed to have mean zero, constant variance and is independent of $X$. Similarly, $q$ is assumed to have a Gaussian distribution with mean zero and constant variance and is independent of $X$ and $U$.

Model (1) above is referred to as *error-in-variable* model, where the covariates are measured with error in a regression setting. Eq.(1a) describes the *classical measurement error* structure, in which only the true (unobserved) covariate $X$ is measured with *additive* error. Eq.(1b) describes the regression structure of the data $Z = \{X, Y\}$. It reduces to the familiar linear regression structure for $g(X; \beta) = X^\top \beta$.

**Remark** *There are other types of error structures such as multiplicative error, but in this paper, we will restrict ourselves to only additive measurement error.*

**Remark** *In measurement error literature, there is an important distinction between the **functional model** where $X$ is not modelled and the **structural model** where $X$ is modelled with a parametric distribution. For this paper, we will restrict our attention to structural modelling where $X$ is assumed to have a Gaussian distribution.*

Under (1) with $g(X; \beta) = X^\top \beta$, one of the most well-known effects of the measurement error is to bias the regression coefficient towards zero. This phenomenon is commonly referred to as *attenuation*. More precisely, the OLS estimator obtained by regressing $Y$ on the surrogate $W$ is not a consistent estimator of $\beta$ but instead of $\beta^* = \lambda\beta$ where

$$\lambda = \frac{\sigma_x^2}{\sigma_x^2 + \sigma_u^2} < 1 \tag{2}$$

The attenuation factor $\lambda$ is referred to as *reliability ratio* (Carroll et al., 2006). The larger the $\sigma_u^2$, the variance of the measurement error, the closer to zero the attenuation factor $\lambda$ will be. Therefore, when ignoring the measurement error, the naive method of regressing $Y$ on $W$ will result in severe underestimation of $\beta$ when the magnitude of measurement error is large.

Based on equation 2, we can obtain a consistent estimator of $\beta$ as

$$\tilde{\beta} = \hat{\beta}_{\text{ols}}/\hat{\lambda} = \hat{\beta}_{\text{ols}} \frac{S_w}{S_w - \hat{\sigma}_u^2} \tag{3}$$

where $S_w$ is the sample variance of the surrogate $W$ and $\hat{\sigma}_u^2$ is a consistent estimator of $\sigma_u^2$.

So far we have only discussed the scenario where only the explanatory variable $X$ is measured with error, but the scenario where both explanatory and response variables are measured with error is much more beneficial to the differential privacy framework. Although there is admittedly less literature on response measurement error, the extension is surprisingly easy in some cases. As noted in Carroll et al. (2006), for unbiased and homoscedastic response measurement error in linear regression, the response measurement error increases the variability of the fitted lines without causing bias. Furthermore, all hypothesis tests, confidence intervals, etc. remain perfectly valid albeit they are less powerful. These conclusions indicate that unbiased error in linear regression requires no special adjustments when extending to response measurement error. However, for the binary response, the measurement error becomes misclassification, which is no longer an unbiased error, and therefore special considerations are required. As the first steps establish the connections between

differential privacy and the measurement error model, we will focus on the linear regression measurement error throughout this paper. Nonlinear regression such as logistic regression will be left for future directions.

# 3 DIFFERENTIAL PRIVACY MECHANISM AS MEASUREMENT ERROR MODEL

## 3.1 GAUSSIAN MECHANISM AS MEASUREMENT ERROR MODEL

Let's denote the private dataset by $\mathbf{Z}$, then the analytic Gaussian mechanism (sec. 2.1) releases a differentially private dataset $\tilde{\mathbf{Z}}$ by adding a centred Gaussian noise $\mathbf{U} \sim \mathcal{N}\left(0, \sigma_u^2 \mathbf{I}_p\right)$,

$$\tilde{\mathbf{Z}} = \mathbf{Z} + \mathbf{U} \tag{4}$$

Note that $\Delta$ denotes the sensitivity for the identity query function, that is, $\Delta := \max_{\mathbf{Z}, \mathbf{Z}'} \|\mathbf{Z} - \mathbf{Z}'\|_F$ where $\|\cdot\|_F$ denotes the Frobenius norm.

Refer back to section 2.2, it's easy to observe that equation 4 can be viewed as the error structure between the surrogate variable $\tilde{\mathbf{Z}}$ and the true unobservable variable $\mathbf{Z}$ of a measurement error model. A key difference here is that commonly in measurement error problems, the magnitude of the measurement error $\sigma_u^2$, the variance of $\mathbf{U}$, is unknown and has to be estimated. Fortunately, in the differential privacy setting, the variance of $\mathbf{U}$ is purely determined by the privacy budget $\varepsilon, \delta$ and the sensitivity $\Delta$, and therefore it can be publicized and is assumed to be known.

**Remark** *For unbounded variables, like Gaussian random variables, $\Delta$ will be $\infty$ and the Gaussian mechanism will no longer work without additional procedures. To deal with unbounded predictors, we simply clip the variable within a fixed interval. To disclose $\Delta$, the fixed interval must be chosen before or without seeing the confidential dataset.*

## 3.2 STATISTICAL INFERENCE FROM MEASUREMENT ERROR PERSPECTIVE

Now equipped with the perspective that the Gaussian mechanism can be viewed as the measurement error structure of a measurement error model, we need the second component of the measurement error model, the data structure, to make statistical inferences.

Let's consider the regression where we partition the private dataset as $\mathbf{Z} = \{\mathbf{X}, \mathbf{y}\}$ where $\mathbf{X}$ is the exploratory variable and $\mathbf{y}$ is the response variable. Furthermore, we assume a functional relationship between $\mathbf{X}$ and the expected value of $\mathbf{y}$,

$$\mathbf{y} = g(\mathbf{X}, \boldsymbol{\beta}) + \mathbf{q} \tag{5}$$

where $\mathbf{q} \sim \mathcal{N}(\mathbf{0}, \sigma_q^2 \mathbf{I})$, and it's assumed to be independent of $\mathbf{X}$.

Combined with equation 4, we can write our measurement error model as the following,

$$\begin{aligned} \mathbf{y} &= g(\mathbf{X}; \boldsymbol{\beta}) + \mathbf{q} \\ \tilde{\mathbf{Z}} &= \mathbf{Z} + \mathbf{U} \end{aligned} \tag{6}$$

where $\tilde{\mathbf{Z}} = (\tilde{\mathbf{X}}, \tilde{\mathbf{y}})$ and $\mathbf{U} = (\mathbf{U}_x, \mathbf{u}_y)$. Note the variance of $\mathbf{U}$ is assumed to be known. When $\mathbf{u}_y$ has a zero variance, then it reduces to model (1), *error-in-variable* model, in section 2.2.

Under model (6), one of the classical methods for estimation is the maximum likelihood approach (Wansbeek & Meijer, 2000) due to several nice properties such as consistency and asymptotic normality that the maximum likelihood estimator (MLE) enjoys. Since only $\tilde{\mathbf{Z}}$ are observed, the likelihood function to maximize comes from the marginal distribution of $\tilde{\mathbf{Z}}$, which is simply a multivariate normal distribution. For some function $g$, numerical analysis is required to maximize the likelihood and a closed-form solution often does not exist. However, in this paper, we will focus on one such scenario that a closed-form solution exists. That is, we will focus on the case that $g(X; \beta) = X^\top \beta$, in which case eq. (6) reduces to the following,

$$\begin{aligned} \mathbf{y} &= \mathbf{X}\boldsymbol{\beta} + \mathbf{q} \\ \tilde{\mathbf{Z}} &= \mathbf{Z} + \mathbf{U} \end{aligned} \tag{7}$$

Denotes [1]

$$\tilde{\boldsymbol{\beta}} = \left(\frac{1}{n}\tilde{\mathbf{X}}^\top\tilde{\mathbf{X}} - \sigma_u^2\mathbf{I}\right)^{-1}\frac{1}{n}\tilde{\mathbf{X}}^\top\tilde{\mathbf{y}}, \quad \tilde{\sigma}_q = S_v - \sigma_u^2\left(1 + \left\|\tilde{\boldsymbol{\beta}}\right\|_2^2\right)$$

where $S_v = \frac{1}{n-k}\left\|\mathbf{y} - \mathbf{X}\tilde{\boldsymbol{\beta}}\right\|_2^2$. To obtain the limiting distribution of these estimators, we have the following theorem.

**Theorem 3.1** (Fuller (1987)). *Let model (7) holds, that is, assume homoscedastic linear regression model, additive measurement error structure and a normally distributed predictor. Let* $\boldsymbol{\theta} = \left(\boldsymbol{\beta}^\top, \sigma_{qq}\right)^\top$ *and let* $\tilde{\boldsymbol{\theta}} = \left(\tilde{\boldsymbol{\beta}}^\top, \tilde{\sigma}_{qq}\right)^\top$. *Then,*

$$n^{1/2}(\tilde{\boldsymbol{\theta}} - \boldsymbol{\theta}) \rightsquigarrow N(\mathbf{0}, \Gamma),$$

*where the submatrices of* $\Gamma$ *are*

$$\Gamma_{\boldsymbol{\beta\beta}} = \mathbf{M}_x^{-1}\sigma_v^2 + \mathbf{M}_x^{-1}\left[\sigma_u^2\sigma_v^2\mathbf{I} + \sigma_u^4\boldsymbol{\beta\beta}^\top\right]\mathbf{M}_x^{-1}$$

$$\Gamma_{qq} = \mathrm{Var}\left(\frac{1}{n}\|\mathbf{v}\|_2^2\right)$$

$$\Gamma_{\boldsymbol{\beta}q} = 2\mathbf{M}_x^{-1}\sigma_u^2\sigma_v^2\boldsymbol{\beta}$$

*with* $\mathbf{v} = \mathbf{u}_y + \mathbf{q} - \mathbf{U}_x\boldsymbol{\beta}$ *and* $\mathbf{M}_x = \boldsymbol{\mu}_x\boldsymbol{\mu}_x^\top + \boldsymbol{\Sigma}_x$. *Furthermore, The variance of the approximate distribution of* $\tilde{\beta}$ *can be estimated by*

$$\widehat{\mathrm{Var}}\{\tilde{\boldsymbol{\beta}}\} = n^{-1}\left[\tilde{\mathbf{M}}_x^{-1}S_v + \tilde{\mathbf{M}}_x^{-1}\left(S_v\sigma_u^2\mathbf{I} + \sigma_u^4\tilde{\boldsymbol{\beta}}\tilde{\boldsymbol{\beta}}^\top\right)\tilde{\mathbf{M}}_x^{-1}\right]$$

*where* $\tilde{\mathbf{M}}_x = \frac{1}{n}\tilde{\mathbf{X}}^\top\tilde{\mathbf{X}} - \sigma_u^2\mathbf{I}$.

**Remark** *The theorem above is not valid if a clipping process is applied to* $\mathbf{Z}$ *to ensure finite sensitivity. Therefore, the clipped interval needs to be sufficiently large to minimize the impact of the clipping effect.*

Directly following the theorem above, we can derive the result for the simple linear regression, $Y = \beta_0 + X\beta_1 + q$, which will be used in the simulation in section 4.

**Corollary 3.1.1** (Simply linear regression (Fuller, 1987)). *Suppose* $\sigma_u^2$ *known,* $\sigma_\varepsilon^2 > 0$, *and* $\sigma_\xi^2 > 0$. *Then, the vector*

$$\sqrt{n}\begin{bmatrix}\tilde{\beta}_0 - \beta_0 \\ \tilde{\beta}_1 - \beta_1\end{bmatrix} \rightsquigarrow \mathcal{N}(\mathbf{0}, \Gamma)$$

*where the covariance matrix* $\Gamma$ *is,*

$$\Gamma = \begin{bmatrix} \mu_x^2\dfrac{\sigma_{\tilde{x}}^2\sigma_v^2 + \mathrm{Cov}^2(\tilde{x}, v)}{\sigma_x^4} + \sigma_v^2 & -\mu_x\dfrac{\sigma_{\tilde{x}}^2\sigma_v^2 + \mathrm{Cov}^2(\tilde{x}, v)}{\sigma_x^4} \\ -\mu_x\dfrac{\sigma_{\tilde{x}}^2\sigma_v^2 + \mathrm{Cov}^2(\tilde{x}, v)}{\sigma_x^4} & \dfrac{\sigma_{\tilde{x}}^2\sigma_v^2 + \mathrm{Cov}^2(\tilde{x}, v)}{\sigma_x^4} \end{bmatrix}$$

*Furthermore,* $n\widehat{\mathrm{Var}}\left\{\left(\tilde{\beta}_0, \tilde{\beta}_1\right)^\top\right\}$ *is a consistent estimator of* $\Gamma$ *where*

$$\widehat{\mathrm{Var}}\left\{\begin{bmatrix}\tilde{\beta}_0 \\ \tilde{\beta}_1\end{bmatrix}\right\} = \begin{bmatrix} \mathrm{mean}(\tilde{X})^2\,\widehat{\mathrm{Var}}\left(\tilde{\beta}_1\right) + \frac{1}{n}S_v & -\mathrm{mean}(\tilde{X})\,\widehat{\mathrm{Var}}\left(\tilde{\beta}_1\right) \\ -\mathrm{mean}(\tilde{X})\,\widehat{\mathrm{Var}}\left(\tilde{\beta}_1\right) & \widehat{\mathrm{Var}}\left(\tilde{\beta}_1\right) \end{bmatrix}$$

$$\widehat{\mathrm{Var}}\left(\tilde{\beta}_1\right) = \frac{1}{n-1}\frac{S_{\tilde{x}}S_v + \tilde{\beta}_1^2\sigma_u^4}{(S_x - \sigma_u^2)^2}$$

*where* $S_v = \frac{1}{(n-2)}\left\|\tilde{\mathbf{y}} - \mathrm{mean}(\tilde{\mathbf{y}}) - \tilde{\beta}_1\left(\mathbf{x} - \mathrm{mean}(\tilde{\mathbf{x}})\right)\right\|_2^2$.

---

[1]Note that $\tilde{\boldsymbol{\beta}}$ is the MLE of $\boldsymbol{\beta}$, but $\tilde{\sigma}_q$ is not the MLE of $\sigma_q$. $\tilde{\sigma}_q$ is used here because the limiting distribution can be derived under less restrictive conditions than those used to obtain the maximum likelihood estimator (Fuller, 1987).

Immediately following from the corollary above, we can derive an asymptotic confidence interval for $\beta_1$ as follows,

**Corollary 3.1.2** (Asymptotic confidence interval). *The interval is defined as follows*

$$\tilde{\beta}_1 \pm t_{1-\alpha/2, n-2} \sqrt{\widehat{\text{Var}}\left(\tilde{\beta}_1\right)}$$

*where $t_{1-\alpha/2, n-2}$ denotes the $1 - \alpha/2$ quantile of the student's t distribution with df $= n - 2$, is a $1 - \alpha$ asymptotically correct confidence interval for the regression coefficient $\beta_1$.*

## 4    SIMULATION AND RESULTS

In this section, we perform simulations to evaluate the performance of our estimator against the widely used SSP algorithm (Dwork et al., 2014; Sheffet, 2017; Wang, 2018; Alabi et al., 2022). As the result will show that our estimator is comparable to the SSP algorithm in some scenarios. Furthermore, we will obtain an asymptotic confidence interval for $\beta_1$ without additional privacy cost, which is one of the advantages of our approach. Compared to the naive CI obtained by ignoring the DP noises, our CI does a much better job capturing the true value for $\beta_1$.

### 4.1    METHOD

For this simulation, we assume the following simple linear regression model,

$$Y_t = \beta_0 + \beta_1 X_t + q_t$$

Additionally, we assume $q_t \sim \mathcal{N}(0, 1)$ and $X_t \sim \mathcal{N}(0, 1)$.

To conduct the simulation, we set the coefficients to be $(\beta_0, \beta_1) = (1, 1)$, and then draw $X_t, t = 1, 2, \ldots, n$ from $\mathcal{N}(0, 1)$ and the regression noises $q_t, t = 1, 2, \ldots, n$ from $\mathcal{N}(0, 1)$. Once $Y_t = \beta_0 + \beta_1 X + q_t, t = 1, 2, \ldots, n$ is obtained, we clip $Y_t$ within the interval $[-3, 3]$ to ensure a finite sensitivity $\Delta$. The particular interval of $[-3, 3]$ is chosen since the interval is relatively large so that the effect of clipping will not have a big impact on the result.

First, we will obtain the point estimators for $\beta_0$ and $\beta_1$ using the SSP algorithm. To implement the SSP algorithm, we follow the `DPSuffStats` algorithm in Alabi et al. (2022) with a few adjustments [2]. To obtain our estimator, we first construct our DP synthetic dataset described in section 3.1 with $\Delta = \sqrt{(1-0)^2 + (3 - (-3))^2} = \sqrt{37}$, and then obtain the estimates as described in section 3.2. To compare the performance between these two estimators, we report their median absolute error (MAE) for each combination of sample size $n \in \{500, 1000, 2000, 5000, 10^4, 10^5\}$ and privacy budget $\varepsilon \in \{0.1, 0.5, 1, 5\}$ while setting $\delta = 1/n$.

Due to the post-processing property of DP, any statistics derived from the DP synthetic dataset will remain differentially private and won't incur any additional privacy risk. Therefore, the asymptotic CI describes in corollary 3.1.2 is differentially private. Similarly, the naive CI obtained by ignoring the DP noises is differentially private as well. To compare our asymptotic CI with the naive CI, we report their relative frequencies of capturing the true value of $\beta_1$ out of the 1000 trials.

Lastly, the normal distribution assumption of the covariates might not be realistic in practice. Therefore, we rerun the simulation described above but with $X_t$ drawn from $\text{Unif}(0, 1)$ instead to evaluate the performance of our method under a different setting.

### 4.2    RESULT

Table 1 shows the MAE results between our DP estimator (bottom value) and the SSP estimator (top value). As we can observe from the table, as one might expect when privacy budget $\varepsilon$ or sample size $n$ increases, the MAE for both estimator decrease. However, the SSP estimator outperforms our SSP estimator except when both sample size and privacy budget are large, their performances are similar. The lower performance of our estimator is due to the nature of the finite sample. In simulation,

---

[2]First, the Gaussian mechanism is used instead of the Laplace mechanism for a better comparison. Then, we extend the algorithm to accommodate different clipping intervals for $X_t$ and $Y_t$.

Table 1: MAE result for uniformly/normally distributed predictor. The top value within each cell indicates the MAE for the SSP algorithm, and the bottom value within each cell indicates the MAE for our estimator without applying the Gram-Schmidt process.

| | Gaussian | | | | Uniform | | | |
|---|---|---|---|---|---|---|---|---|
| | $\varepsilon = 0.1$ | $\varepsilon = 0.5$ | $\varepsilon = 1$ | $\varepsilon = 5$ | $\varepsilon = 0.1$ | $\varepsilon = 0.5$ | $\varepsilon = 1$ | $\varepsilon = 5$ |
| $n = 500$ | 2.139 | 0.592 | 0.342 | 0.148 | 6.497 | 1.884 | 1.040 | 0.301 |
| | 7.105 | 2.871 | 2.029 | 1.117 | 5.400 | 2.504 | 2.039 | 1.757 |
| $n = 1000$ | 1.218 | 0.334 | 0.205 | 0.132 | 3.938 | 1.001 | 0.568 | 0.183 |
| | 5.929 | 2.476 | 1.862 | 1.018 | 4.557 | 2.310 | 1.889 | 1.735 |
| $n = 2000$ | 0.699 | 0.200 | 0.140 | 0.127 | 2.212 | 0.557 | 0.307 | 0.121 |
| | 5.026 | 2.185 | 1.652 | 0.838 | 3.963 | 2.114 | 1.837 | 1.652 |
| $n = 5000$ | 0.324 | 0.131 | 0.127 | 0.128 | 0.984 | 0.244 | 0.147 | 0.090 |
| | 3.976 | 1.876 | 1.508 | 0.681 | 3.218 | 1.893 | 1.801 | 1.596 |
| $n = 10^4$ | 0.198 | 0.128 | 0.129 | 0.128 | 0.551 | 0.138 | 0.097 | 0.080 |
| | 3.401 | 1.711 | 1.392 | 0.574 | 2.858 | 1.815 | 1.805 | 1.481 |
| $n = 10^5$ | 0.128 | 0.128 | 0.128 | 0.128 | 0.088 | 0.075 | 0.074 | 0.074 |
| | 2.048 | 1.339 | 1.111 | 0.276 | 1.978 | 1.803 | 1.726 | 1.151 |

the sample covariance between DP noises and data is often non-negligible when the sample size is small relative to the amount of noises injected. This results in poor estimation of $\sigma_x$, which leads to the poor performance of our estimator. Although our estimation performance is worse than the SSP method, it's still comparable in some scenarios (the combination of a small privacy budget and a small sample size or the combination of a large privacy budget and a large sample size). Furthermore, our approach allows the release of a synthetic dataset and more importantly, it provides the method to obtain a confidence interval without additional privacy budget. We will discuss the performance of our confidence interval next.

Figure 1 show the coverage probabilities and margin of error of our confidence intervals (DP) under normally distributed and uniformly distributed covariate $X$. For comparison, the naive CIs derived from the synthetic dataset (naive) and non-DP CIs (non-DP) derived from the confidential dataset are also plotted. As shown in both figures, the coverage of our CIs is relatively close to the nominal level (90%, indicated by the dotted lines). In comparison, the coverage of the naive CIs never captures the true value of $\beta_1$ even though they are much narrower in comparison. This highlights the importance of considering DP noises when making statistically valid inferences. The reason behind the terrible coverage of the naive CIs, as explained in Carroll et al. (2006), is because the variance of the naive estimator can be smaller than the true data estimator when the privacy budget is small (or DP noises are large), which results in a more precise, but biased estimator.

## 5 CONCLUSION

In this paper, we established a connection between DP mechanisms and measurement error models. Applying the tools from the measurement error framework, we developed statistical inference under linear regression while preserving differential privacy. In particular, we derived DP consistent estimator and DP asymptotic conference interval for the regression coefficients. To evaluate the performance of the estimator, we compared it to the widely used SSP method and demonstrated our estimator has comparable performance in some scenarios but has the advantage of obtaining statistically valid asymptotic confidence intervals without additional privacy cost. Furthermore, by comparing the coverage between our asymptotic CIs and naive CIs, we illustrated the importance of incorporating the DP mechanism into the inference method to ensure a valid statistical inference.

For future directions, some theoretical works on the comparison between our estimator and the SSP estimator could be an interesting direction. Similarly, the extension of the theorem 3.1 to accommodate the clipping might be a fruitful path to pursue. Furthermore, there are still many tools from the measurement error literature yet to be utilized. One of the obvious next steps would be to extend the linear regression setting to the more general generalized linear model setting such as logistic regression. We hope this paper will motivate future works to explore more the connection

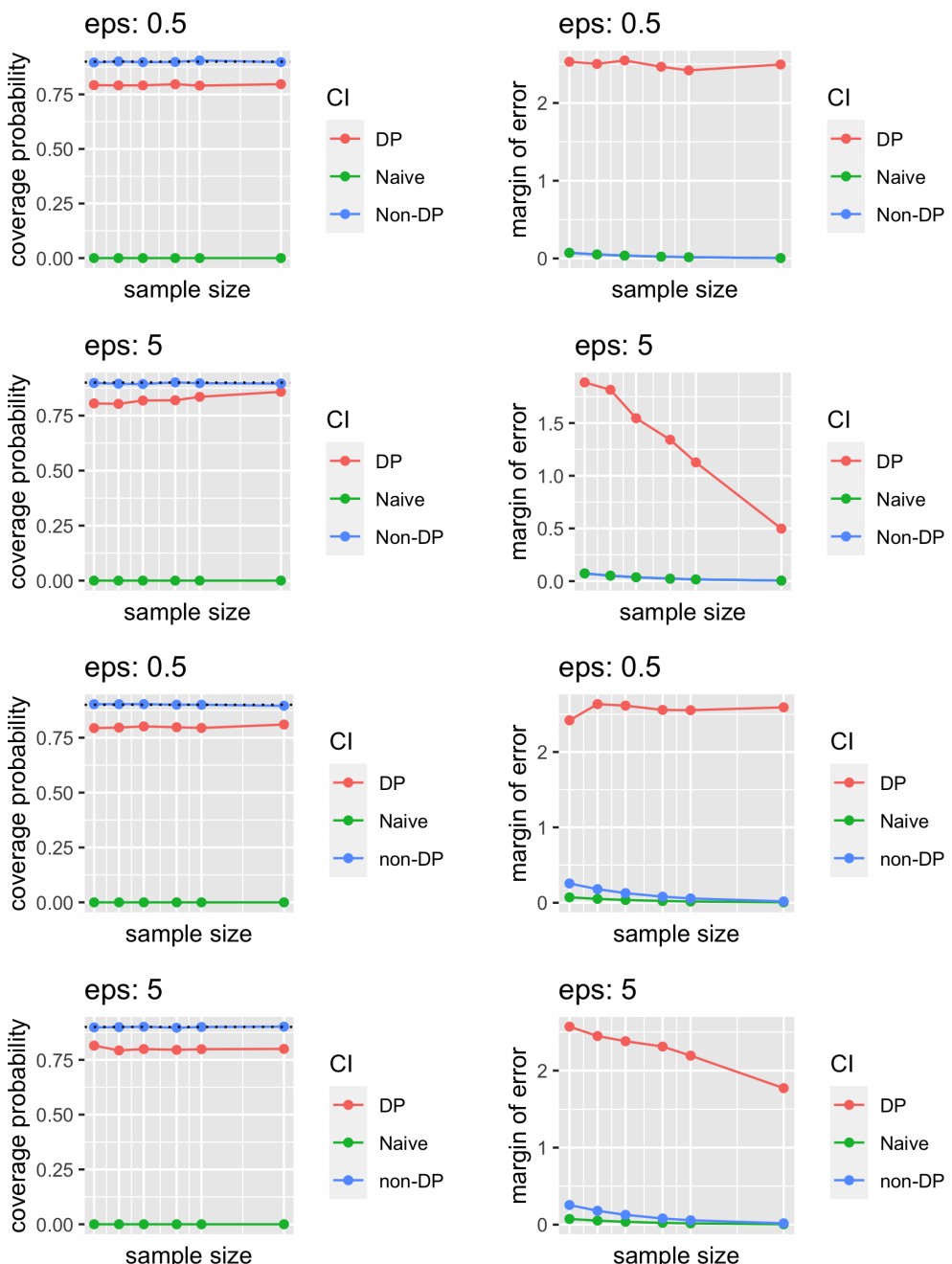

Figure 1: The coverage probabilities comparison between our DP confidence intervals, naive CIs and non-DP CIs at various sample sizes. The top 4 plots are for the normally distributed covariate, and the bottom 4 plots are for the uniformly distributed covariate. Note the horizontal axis is in logarithmic scale with sample size $n = 500, 1000, 2000, 5000, 10^4, 10^5$. The dotted line indicates the nominal CI level of $0.9$.

between differential privacy and measurement error, and to develop statistical inference under the differential privacy setting.

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
