# OpenReview forum: "Analysis of differentially private synthetic data: a general measurement error approach"
_ICLR.cc/2023/Conference — Submitted to ICLR 2023_

### Official Review · Reviewer_cuVk · 2022-10-24

**Confidence:** 3
**Correctness:** 4
**Technical Novelty And Significance:** 4
**Empirical Novelty And Significance:** Not applicable
**Recommendation:** 5

**Clarity, Quality, Novelty And Reproducibility:**

The paper is relatively clear and well-written, and as far as I know novel. With the information available in the paper, I think that the experiments are reproducible although I haven't done it.

**Strength And Weaknesses:**

Strengths
- The paper tackles an interesting problem, namely statistical inference using DP synthetic datasets
- The paper makes the connection to measurement error models which is to the best of my knowledge new

Weaknesses
- The proposed estimator seems to be worse than the baseline in practical scenarios


Typos
- The title ends with "conference submissions"
- Missing \ref in Definition 2.4

**Summary Of The Paper:**

The paper tackles the problem of statistical inferences from differential private synthetic datasets. The authors use the tools of measurement error models to derive an estimator that is bias-corrected. Experiments are performed that compare the proposed estimator to SSP (sufficient statistic perturbation), which show that the superiority of SSP in practice, but a better performance of the proposed estimator if the sample covariance was 0.

**Summary Of The Review:**

The paper proposes a connection between DP and measurement error models. While I think it is a new connection, the results seem limited and underperform the proposed baseline. I am not knowledgeable about measurement error models so I might miss some aspect of it, but overall I would lean towards reject.

---

> ### Author Response · Authors · 2022-11-18
> **Rebuttal Answer**
>
> Thank you for your review and excellent feedback. We address your concerns below.
>
> > The proposed estimator seems to be worse than the baseline in practical scenarios
>
> Our estimator indeed performs worse than the baseline estimator (SSP estimator) on the estimation of the regression coefficients. However, our approach allows us to obtain additional statistics such as confidence intervals without additional privacy risk, which cannot be said for the baseline approach.
>
> **Typos**: Thank you for pointing it out. The typos have been fixed.

---

### Official Review · Reviewer_3s3F · 2022-10-25

**Confidence:** 4
**Correctness:** 3
**Technical Novelty And Significance:** 2
**Empirical Novelty And Significance:** Not applicable
**Recommendation:** 5

**Clarity, Quality, Novelty And Reproducibility:**

Clarity

1) Please completely state the assumptions of Theorem 3.1.

2) Please streamline the model description in section 2. It is currently very spread out, with detours to describe models that are not actually analyzed (nonlinearity etc).

3) Please fix various grammatical issues, listed below
-missing space after “))” in section 1
-“reinsurance” in section 2
-analytic Gaussian mechanism citation missing in section 2
-“now equipped with the respect” in section 3
-“immediate follow from…” in section 3

Quality

1) A full statement of the assumptions of Theorem 3.1 is important to establish soundness.

2) There are some discrepancies between theory and practice which I would like to be addressed.
-If I understand correctly, the theory requires X to be Gaussian while the simulations take X to be uniform
-To get good performance, the authors impose Gram-Schmidt orthogonalization as a kind of pre-processing step, which is not theoretically justified

3) I found the promise of a “general class of bias-corrected DP estimators’’ in the abstract to be an overstatement; there is one estimator that inverts the well-known attenuation bias formula for regression.

Novelty

My primary dissatisfaction is the framing of the stated contributions in the context of previous works which make one or both of these points.

1) The connection between measurement error models and additive differential privacy mechanisms. The authors mention some papers, but there are several more, e.g. Agarwal et al. (NeurIPS 2019, JASA 2021) and papers that build on it.

2) Using this connection to provide inference results to the differential privacy setting.  Agarwal and Singh (arXiv 2021) use this connection to prove inference results for regression coefficients as well as more general quantities, without requiring linearity and homoscedastic Gaussians. Relative to that work, the appropriate framing is perhaps that when the researcher has much more structure on the problem (e.g. linearity, homoscedasticity, Gaussianity), and the goal is to estimate a regression coefficient, then a simple inversion of the attenuation bias formula is enough.

**Strength And Weaknesses:**

The strength is that the proposed solution is very simple. The weaknesses are described as actionable items below

**Summary Of The Paper:**

The authors consider the standard error-in-variables model: Y=X’b+q and tilde{X}=X+U. The authors impose that X, U, and q are homoscedastic Gaussians with fixed dimensions. Then the standard attenuation bias formula can be inverted to correct for the attenuation bias, if the variance of q is known. The authors quote estimation and inference results from Fuller (1987). The stated contribution is twofold:
1) establishing the connection between measurement error models and additive differential privacy mechanisms
2) using this connection to repurpose known inference results to the differential privacy setting

My primary dissatisfaction is the framing of these results in the context of previous works which make one or both of these points.

**Summary Of The Review:**

The main result is essentially an interpretation of the standard attenuation bias formula. The shortcomings in terms of the clarity, the gap between theory and practice, and the inadequate framing limit the significance of the contribution. I will improve the score if these shortcomings are addressed.

---

> ### Author Response · Authors · 2022-11-18
> **Rebuttal Answer**
>
> Thank you for your review and excellent feedback. We address your concerns below.
>
> **Clarity**:
>
> 1, 2 & 3: Thank you for pointing it out. The typos have been fixed. Section 2.2 has been rewritten for better readability.
>
> **Quality:**
>
> The full assumption has been added in the theorem.
>
> > There are some discrepancies between theory and practice which I would like to be addressed. -If I understand correctly, the theory requires X to be Gaussian while the simulations take X to be uniform -To get good performance, the authors impose Gram-Schmidt orthogonalization as a kind of pre-processing step, which is not theoretically justified
>
> We used uniform distribution for the covariate because we want to reduce the impact of clipping. We added the result for normally distributed covariates in the revision. The result assuming orthogonal DP noises is only included to demonstrate the "best case scenario" performance of our estimator. This assumption is unreasonable and unlikely to hold in practice, thus we have removed the result to avoid confusion.
>
> > I found the promise of a “general class of bias-corrected DP estimators’’ in the abstract to be an overstatement; there is one estimator that inverts the well-known attenuation bias formula for regression.
>
> Yes, we agree that a "general" class seems to be an overstatement as we are only focusing on the Gaussian design setting. The corresponding adjustments have been made to reflect that. On the other hand, our method can be extended to more general settings other than $g(X; \beta) = X^\top \beta$ through the maximum likelihood approach.
>
> **Novelty:**
>
> > The connection between measurement error models and additive differential privacy mechanisms. The authors mention some papers, but there are several more, e.g. Agarwal et al. (NeurIPS 2019, JASA 2021) and papers that build on it.
>
> More papers mentioning the connection have been added.
>
> > Using this connection to provide inference results to the differential privacy setting. Agarwal and Singh (arXiv 2021) use this connection to prove inference results for regression coefficients as well as more general quantities, without requiring linearity and homoscedastic Gaussians. Relative to that work, the appropriate framing is perhaps that when the researcher has much more structure on the problem (e.g. linearity, homoscedasticity, Gaussianity), and the goal is to estimate a regression coefficient, then a simple inversion of the attenuation bias formula is enough.
>
> More papers using the connection for statistical inference have been added. Thanks for the suggestion, and we have added that although our setting is not as general as Agarwal and Singh (arXiv 2021), our approach gives a more simplistic solution when there are more structures.

---

> > ### Comment · Reviewer_3s3F · 2022-12-05
> > **Thank you for these improvements**
> >
> > I thank the authors for these improvements, and I have raised the score

---

### Official Review · Reviewer_ffBc · 2022-10-26

**Confidence:** 3
**Clarity, Quality, Novelty And Reproducibility:** paper is clearly written and is easy …
**Correctness:** 3
**Technical Novelty And Significance:** 3
**Empirical Novelty And Significance:** 3
**Recommendation:** 5

**Strength And Weaknesses:**

The proposed method draws interesting parallels between the measurement error theory and DP synthetic data and shows how one can account for the other to generate good estimators. I like the idea, but I think the limitations of the proposed method need more discussion. Such as, how applicable is it in the real world? especially when the input data to a regression model does not follow any nice properties as ensured by simulations. What happens when sensitivity is large (i.e. it is not possible to constrain data in [0,1] without destroying utility)?

I do understand that the proposed method is claimed to be the first to draw this connection, so empirical evaluation comes in second, but I do think it requires more extensive experimentation. Especially, comparing (either via experiments or by including a good discussion), on the advantage of using this compared to generating synthetic data using say TableGAN, and then doing a normal regression.

**Summary Of The Paper:**

Paper uses the DP synthetic data, generated by adding Gaussian noise (Gaussian mechanism) to the sensitive data, to model the analysis using the measurement error approach. Simulations show that the proposed method works well for linear regression.

**Summary Of The Review:**

I hope the authors can answer my questions regrading the implications in real world scenarios, I am open to increasing my score after rebuttal.

---

> ### Author Response · Authors · 2022-11-18
> **Rebuttal Answer**
>
> Thank you for your review and excellent feedback. We address your concerns below.
>
> > What happens when sensitivity is large?
>
> The covariates do not need to be constrained within $[0, 1]$. In the simulation, we used $\text{Unif}(0, 1)$ for the covariate because we want to reduce the impact of clipping. We added the result for normally distributed covariates (clipped within the [-3, 3] interval to ensure finite sensitivity) in the revision, and the results are similar.
>
> > Especially, comparing (either via experiments or by including a good discussion), on the advantage of using this compared to generating synthetic data using say TableGAN, and then doing a normal regression.
>
> We choose to generate synthetic datasets by adding gaussian noises because of its simplicity. Due to its simplicity, we can easily incorporate the synthetic dataset generation process into our model. This way, the synthetic dataset generation is considered when making statistical inferences using the synthetic dataset. If generating synthetic datasets using a method like TableGAN, the dataset generation process could be difficult to incorporate into the model and doing regression using the synthetic dataset without accounting for the generation process will lead to an invalid inference.

---

### Official Review · Reviewer_cBN5 · 2022-10-28

**Confidence:** 4
**Correctness:** 3
**Technical Novelty And Significance:** 2
**Empirical Novelty And Significance:** 2
**Recommendation:** 3

**Clarity, Quality, Novelty And Reproducibility:**

There are some other issues that could be easily addressed:
- Section 2.1: in the discussion of differential privacy, it is mentioned that there is no "additional privacy risk" for individuals whether their data belongs to the private dataset or not. This should be rephrased. The goal of DP is not to have zero risk, but rather to have a bounded risk.
- In Definition 2.4: there is a broken reference, and $\Phi$ is not defined.
- The captions for Tables 1-2 seems to be reversed.
- Regarding the experiments: Section 4.1 assumes that $X$ is uniform in (0, 1), but the analysis of Sections 2-3 assumes a Gaussian design matrix. Why this discrepancy?
- Can you clarify if the the theoretical results (Theorem 3.1 and Corollary 3.1.1) are valid under clipping?

**Strength And Weaknesses:**

## Strengths
- The paper is clearly written.
- Empirical experiments show an improvement in terms of estimating confidence intervals.

## Weaknesses
I primarily have the following questions/concerns:
1) The empirical study suggests that there is an improvement in average error only when the covariance matrices of the data and the added noise are orthogonal. However, this does not seem to be a realistic scenario, and I fail to see what conclusion is drawn from this experiment. Can you clarify why it's reasonable to assume the covariance matrices to be orthogonal?
2) There is no theoretical comparison to prior methods in terms empirical or population risk. In particular, different variants of the sufficient statistics perturbation method have been studied under different scenarios (Gaussian design, fixed, bounded and unbounded designs). It will be important to understand how the proposed method compares to existing methods.

**Summary Of The Paper:**

The paper uses a connection between differentially private linear regression and statistical inference under measurement error. The main technical contribution is to use this connection to apply prior results from the measurement error model to DP linear regression.

**Summary Of The Review:**

The contribution of the paper is to apply prior results from the measurement error model to differentially private linear regression, so it does not introduce new tools, techniques, or proofs. The empirical results are unconvincing due to the unrealistic assumption on orthogonality of noise/data covariances.

---

> ### Author Response · Authors · 2022-11-18
> **Rebuttal Answer**
>
> Thank you for your review and excellent feedback. We address your concerns below.
>
> **Weaknesses**:
>
> > The empirical study suggests that there is an improvement in average error only when the covariance matrices of the data and the added noise are orthogonal. However, this does not seem to be a realistic scenario, and I fail to see what conclusion is drawn from this experiment. Can you clarify why it's reasonable to assume the covariance matrices to be orthogonal?
>
> The result assuming orthogonal DP noises is only included to demonstrate the "best case scenario" performance of our estimator. This assumption is unreasonable and unlikely to hold in practice. Thus, we have removed the result to avoid confusion. The key takeaway we hope to achieve is that our estimator has comparable performance to the SSP estimator in some scenarios but has the advantage of obtaining additional statistics such as confidence interval without additional privacy cost.
>
> > There is no theoretical comparison to prior methods in terms empirical or population risk. In particular, different variants of the sufficient statistics perturbation method have been studied under different scenarios (Gaussian design, fixed, bounded and unbounded designs). It will be important to understand how the proposed method compares to existing methods.
>
> We agree that the lack of theoretical comparison is a point of weakness. However, the two simulations (one simulation for uniformly distributed covariate, and one for normally distributed covariate) should provide some comparisons between our estimator and the SSP estimator in different settings.
>
> **Clarity, Quality, Novelty And Reproducibility**:
>
> 1, 2 & 3: Thank you for pointing it out. The typos have been fixed. Indeed, differential privacy only ensures bounded risk. This has been corrected in the revision.
>
> > Regarding the experiments: Section 4.1 assumes that  X is uniform in (0, 1), but the analysis of Sections 2-3 assumes a Gaussian design matrix. Why this discrepancy?
>
> We used uniform distribution for the covariate because we want to reduce the impact of clipping. We added the result for normally distributed covariates in the revision.
>
> > Can you clarify if the the theoretical results (Theorem 3.1 and Corollary 3.1.1) are valid under clipping?
>
> The theoretical results are not valid under clipping. Therefore, the interval used for clipping should be sufficiently large so that the impact of clipping is minimized. This has been emphasized in the revision.

---

### Decision · Program_Chairs · 2023-01-20

**Decision:**

Reject

**Justification For Why Not Higher Score:**

There were a variety of concerns regarding soundness and applicability (both as is and in more general setups.

**Justification For Why Not Lower Score:**

N/A

**Metareview: Summary, Strengths And Weaknesses:**

The paper introduces a view of differential privacy that is linked to the literature on measurement error models, which allows for some fruitful exchange of ideas.

Strengths: simplicity of idea; the connection described appears sounds and can suggest new ideas to be pursued;

Weaknesses: there were a variety of concerns regarding soundness and applicability (both as is and in more general setups).